# Type 1 dopamine receptor (D1R)-independent circadian food anticipatory activity in mice

**Dina R. Assali, Michael Sidikpramana, Andrew P. Villa, Jeffrey Falkenstein, Andrew D. Steele** *

Department of Biological Sciences, California State Polytechnic University Pomona, Pomona, CA, United States of America

* adsteele@cpp.edu

**Data Availability Statement:** All relevant data are within the paper and its Supporting Information files.

## Abstract

Circadian rhythms are entrained by light and influenced by non-photic stimuli, such as feeding. The activity preceding scheduled mealtimes, food anticipatory activity (FAA), is elicited in rodents fed a limited amount at scheduled times. FAA is thought to be the output of an unidentified food entrained oscillator. Previous studies, using gene deletion and receptor pharmacology, implicated dopamine type receptor 1 (D1R) signaling in the dorsal striatum as necessary for FAA in mice. To further understand the role of D1R in promoting FAA, we utilized the Cre-lox system to create cell type-specific deletions of D1R, conditionally deleting D1R in GABA neurons using *Vgat-ires-Cre* line. This conditional deletion mutant had attenuated FAA, but the amount was higher than expected based on prior results using a constitutive knockout of D1R, *D1R KO^Drago*. This result prompted us to re-test the original *D1R KO^Drago* line, which expressed less FAA than controls, but only moderately so. To determine if genetic drift had diminished the effect of D1R deletion on FAA, we re-established the *D1R KO^Drago* knockout line from cryopreserved samples. The reestablished *D1R KO^Drago-cryo* had a clear impairment of FAA compared to controls, but still developed increased activity preceding mealtime across the 4 weeks of timed feeding. Finally, we tested a different deletion allele of *D1R* created by the Knockout Mouse Project. This line of *D1R KO^KOMP* mice had a significant impairment in the acquisition of FAA, but eventually reached similar levels of premeal activity compared to controls after 4 weeks of timed feeding. Taken together, our results suggest that D1R signaling promotes FAA, but other dopamine receptors likely contribute to FAA given that mice lacking the D1 receptor still retain some FAA.

## Introduction

Circadian rhythms link molecular processes within cells to large scale phenomena within organisms such as metabolism, hormonal signaling, and behavior [1–3]. The circadian time-keeping mechanism for photoentrainment has been linked to core transcriptional-translational feedback loops in many types of cells [4]. In mammals, the suprachiasmatic nucleus (SCN) serves as the main time keeper, orchestrating the 24 hour rhythms in many other brain

**Funding:** The Whitehall Foundation provided a Research Grant to AS, and the National Institute of General Medical Sciences of the National Institutes of Health provided funding under Award Number SC3GM125570 to AS. The content is solely the responsibility of the authors and does not necessarily represent the official views of the National Institutes of Health. The funders had no role in study design, data collection and analysis, decision to publish, or preparation of the manuscript.

**Competing interests:** The authors have declared that no competing interests exist.

regions and peripheral organs [5]. Through photoreceptors within the retina, the SCN is synchronized by daily light exposure [6,7].

Aside from photic input, the availability of food is an external cue that synchronizes circadian clocks in the brain and peripheral tissues independently of the SCN [8]. This is demonstrated in rodents with time-restricted food access during their rest phase, which causes a shift of circadian activity cycles while retaining nocturnal activity driven by the SCN. This shift is associated with a bout of locomotor activity that anticipates the daily scheduled meal by about two hours [9]. Remarkably, this food anticipatory activity (FAA) is robustly maintained in rodents with SCN lesions, indicating the presence of a food entrained oscillator separate from the SCN [10]. It is not clear if the molecular components of the food entrained oscillator are fully, partially, or non-overlapping with the light entrained oscillator present within the SCN [11–13]. The location of the neurons responsible for the anticipation of food and other related stimuli have been elusive [9,14–17]. Recent studies have implicated everything from liver-derived ketones (requiring intact period gene in the liver) [18] to neuronal expression of Reverb-α (Nr1d1) [19] to dopamine receptors 1 and 2 as important for mediating FAA [20–22].

Within the brain, dopamine signaling is essential for orchestrating movement and plays a major role in reward and motivation [23–25]. The dopamine neurons in the substantia nigra (SN) and ventral tegmental area (VTA) innervate several parts of the brain, with SN neurons projecting primarily to the dorsal striatum which regulates goal-directed locomotion, while VTA neurons project to the nucleus accumbens to drive motivation and learning [25]. Studies have shown functional overlap and interactions between the SN→dorsal striatum and VTA→nucleus accumbens dopamine circuits [26,27]. Dopamine neurons function by activating two distinct types of G-protein coupled receptors, D1-like and D2-like, which are expressed in abundant, intermingled populations in the striatum and nucleus accumbens, among other regions [28,29]. Activation of D1-like receptors increases intracellular cAMP production to increase neuronal activity and excitability while D2-like receptors function to lower the intracellular production of cAMP, thereby depressing activity in D2-expressing neurons [29].

Dopamine signaling is modulated by circadian machinery and, conversely, dopamine signaling influences circadian rhythms [20–22,30–35]. For example, Reverb-α, a circadian nuclear receptor, is a negative regulator of tyrosine hydroxylase (TH) mRNA levels; TH levels were found to be highest at night when REV-ERBα levels were lowest [34]. In addition, mice with *REV-ERBα* gene deletion had higher dopamine production and firing rates than did controls [34]. Another study reports that extracellular levels of dopamine are controlled by the dopamine transporter protein (*DAT* or *Slc6a3*) with levels of dopamine correlating the behavioral activity cycle [32]. Furthermore, Hood and colleagues demonstrated that period 2 (Per2) protein expression in the striatum follows a circadian pattern and is dependent on dopamine signaling, as rats with unilateral lesions to dopamine neurons lost their Per2 rhythm in the lesioned side but not contralateral side [31]. Therefore, further understanding of the timing and release of dopamine is necessary for the treatment of many disorders, including drug addiction [36].

Daily access to food leads to anticipatory behavioral activity that may also be dependent on dopamine signaling [30,37,38]. Previous studies of mice with a constitutive knockout (KO) of D1R showed attenuated FAA, whereas D2R KO mice showed normal FAA [21,39]. The D1R KO mice had a weakened *PER2* RNA expression in their striatum as well. In mice lacking TH, viral reintroduction of dopamine production in the dorsal striatum was sufficient for FAA, suggesting that dopamine signaling to D1R neurons in the dorsal striatum permits the expression of FAA [21]. In addition, timed daily D1R agonist injection caused anticipatory activity

even in mice fed *ad libitum* (AL), suggesting that D1R neurons can control behavioral rhythms [21]. Taken together, these results suggested that the SN dopamine-striatal D1R neurons may constitute a critical component of the circadian response to scheduled feeding. Thus, resolving the D1R-dopamine circuitry required for FAA would advance the field toward a circuit-level understanding of food as a zeitgeber. To that end we examined conditional deletion mutants of D1R and also re-examined constitutive D1R mutants.

## Materials and methods

### Mouse husbandry and strains

The Institutional Animal Care and Use Committee at the California State Polytechnic University Pomona approved the experiments described herein. Mice were maintained on a 12:12 light:dark cycle in static microisolator cages. By convention, zeitgeber time (ZT) 0 was defined as lights on, while ZT 12 was designated as lights-off. Home cages contained sani-chip bedding (Envigo, 7090) and a cotton nestlet. Temperatures ranged between 22-24°C and humidity between 20–45%. For all experiments, except those described in Fig 2 with D1R KO$^{Drago}$, mice were fed rodent chow (2018 Teklad Diet Envigo), which has a caloric density of 3.1 kcal/gram and macronutrient composition of 18% fat, 24% protein, 58% carbohydrates. For the experiments with D1R KO$^{Drago}$ mice, n = 10 control and n = 10 KO were fed formulated chow (Research Diets, D12450K), which had a caloric density of 5.49 kcal/gram and a macronutrient composition of 10% fat, 20% protein, 70% carbohydrates. For this experiment we combined data from n = 9 controls and n = 13 D1R KO$^{Drago}$ fed standard chow with those fed unformulated chow since both diets yielded similar results on feeding, body weight, activity, and FAA.

*Drd1$^{tm1e(KOMP)Wtsi}$* mice (Denoted as *D1R KO$^{KOMP}$*) were obtained from the KOMP repository [40], whereas Cre-driver mice were obtained from the Jackson Laboratory (Bar Harbor, ME) (Table 1). Drd1$^{tm1Jcd}$/J heterozygote (+/-) mice (Denoted as *D1R KO$^{Drago}$*) were bred for 3–5 generations before being tested for FAA, with only WT/WT (+/+) and KO/KO (-/-) mice being used for CR experiments. Tg(Drd1a-Cre)$^{120MXu}$ (denoted as *D1-Cre*) and Slc32a1$^{tm2(cre)Lowl}$ (denoted as *Vgat-Cre*) were crossed with mouse strain Drd1$^{tm2.1Stl}$/J, which contain LoxP flanked *D1R* (denoted as *D1R$^{flox/flox}$*; these mice were kindly provided by Dr. Susumu Tonegawa [MIT] and are now available from Jackson Labs stock 025700). For CR experiments with Drd1$^{tm1e(KOMP)Wtsi}$ (Denoted as *D1R KO$^{KOMP}$*) and *Vgat-Cre* mice, both homozygous wild-type and heterozygous wild-types were used as controls. Both males and females were used for all studies.

To genotype mice, DNA was obtained from tail clippings from 2-week old mice, which were then digested with proteinase K and DNA was purified by isopropanol precipitation. Genomic PCR was performed to amplify D1R and Cre alleles. The *D1R KO$^{KOMP}$* mice were genotyped through a third party using qPCR (Transnetyx). Genomic DNA of the *D1R KO$^{Drago}$* and *D1R KO$^{KOMP}$* lines was assessed using single polymorphic nucleotides (SNPs) for differences between genetic background comparable to C57BL/6J and C57BL/6N WT mice (Table 2) by the Jackson Laboratory.

### Food intake, behavioral measurements and calorie restriction conditions

Food intake was measured at 9–10 weeks of age. To calculate 60% CR food allotment per cohort, daily food intakes were measured per mouse over 48 hours. These values were then averaged per group and 60% of the daily average was determined as the daily CR value. Food intakes were adjusted by 0.1 g per day to prevent excessive weight loss for some individual mice (experimenters were blind to genotypes during the experiment).

On video recording days, home cages only contained a minimum of sani-chip bedding (100ml) and no cotton nestlet. During video recording, dim red lighting was provided with High Power 42 SMT RED LED PAR38 from LEDwholesalers.com. Home cage behavior measurements were obtained by video recording mice from a perpendicular angle to their home cages and analyzing these videos using an automated-behavior recognition software, HomeCageScan 3.0 [41], which annotates for the following behaviors: remain low, pause, twitch, awaken, distance traveled, turn, sniff, groom, food bin entry, chew, drink, stretch, hanging, jumping, rearing, walking, and unassigned behaviors. These data were binned in 24 one-hour bins to quantify the temporal structure of activity. The behaviors hanging, jumping, rearing, and walking are designated as high activity. Total high activity was then determined by summation of high activity bins, while FAA ratios were calculated by dividing the final 3h of high activity over the total high activity of each mouse.

All mice entering CR were at least 10 weeks of age. Mice were individually housed on AL for 5–7 days prior to beginning CR with food intake studies conducted at 2–4 days preceding CR. Mice were then fed 60% of their daily caloric intake every day for 28 days at ZT 6 beginning on Day 0, or the start of CR. Body weights were measured every 7th day prior to scheduled mealtime, beginning from Day -7 and ending at Day 28.

### Tissue histology and antibody labeling

For immunolabeling, mice were euthanized with $CO_2$ gas and their brains were perfusion-fixed using 4% paraformaldehyde (Sigma) and post-fixed for 24h before being sectioned at room temperature using a vibrating blade microtome (VT-1000S, Leica Instruments). Antibody staining was performed using a rat monoclonal D1R antibody 1:250 (Sigma) and a chicken polyclonal tyrosine hydroxylase (TH) 1:500 (Aves). An Alexa Fluor 647-conjugated AffiniPure Goat Anti-Chicken IgY 1:500 and an AffiniPure Goat Anti-Rat 488 1:500 were used as fluorotags, along with DAPI (4', 6-Diamidino-2-Phenylindole, Dihydrochloride) 1:1000 for nuclear staining. Sections were imaged by confocal microscopy using a Nikon Eclipse Ti-E inverted microscope system linked to a computer system with NIS-Elements Imaging Software and FIJI was used for image processing [42].

### Statistical analysis

Behavioral data were exported from HomeCageScan 3.0 as excel files, which were analyzed using MATLAB programs to sum, average, and visualize data. Statistical tests and graphs for all data described were performed using GraphPad Prism 9. Sample sizes for each experiment are indicated in the figure legends. Food intake data, which fell on a normal distribution, were tested for significance using an Unpaired T Test. Body weights were analyzed for differences using mixed-effects analysis with Tukey's post-test to examine differences between groups at each time point. For home-cage behavioral data, we also used mixed-effects analysis to examine the effects of genotype, time, and the interaction between time and genotye. We used Sidak's multiple comparisons test to compare behavior at each time point. To analyze a contingency table for mouse genotypes, a Chi-Square Test was performed to determine the dependency of genotype on viable offspring.

## Results

### Conditional deletion of D1R using D1R-Cre may be embryonically lethal

To follow up on our previous work, which utilized a constitutive deletion mutant of D1R (denoted as *D1R KO^{Drago}*), we attempted to conditionally delete *D1R* by crossing *D1R^{flox/flox}*

mice [43] to *D1-Cre* transgenic mice [44] (Table 1). Surprisingly, backcrossing F$_1$ progeny (*D1-Cre$^{+/-}$; D1R$^{wt/flox}$*,) to the parental *D1R$^{flox/flox}$* line did not produce cKO progeny (*D1-Cre$^{+/-}$,D1R$^{flox/flox}$*): out of 47 progeny examined, all were negative for this genotype (Table 3; P<0.0001, Chi-Square). These results suggest that conditionally deleting *D1R* in all *D1R-Cre* expressing cells is embryonically lethal. However, since the location of the *D1R-Cre* transgene was not determined, the lack of cKO progeny can also be attributable to a genetic linkage of the *D1R-Cre* transgene with the *D1R* locus. Nevertheless, the focus of our investigation does not concern the cause of embryonic lethality and therefore we turned to a different *Cre* driver line to achieve deletion of *D1R*.

## Conditional deletion of D1R in GABAergic neurons using Vgat-Cre reduces food anticipatory activity

A second attempt at a broad conditional deletion of *D1R* was made using a well-characterized *Cre* driver for inhibitory neurons, *Vgat-Cre* [45]. The only known D1R-expressing neurons in the dorsolateral striatum are GABAergic medium-sized spiny neurons [46]; thus, we expected that *D1R* should be deleted throughout the striatum in *Vgat-Cre; D1R$^{flox/flox}$* mice. We verified deletion of D1R protein using antibody labeling. Both D1R and TH staining can be seen in control striatum tissue (Fig 1A, 1C and 1E), whereas D1R antibody labeling is completely absent in *Vgat-Cre; D1R$^{flox/flox}$* forebrain tissue (Fig 1B, 1D and 1F).

We measured daily food intake levels for control and *Vgat-Cre; D1R$^{flox/flox}$* mice, and found no difference in mean food intakes: control animals ate 4.2g (+/-0.6g SD) and *Vgat-Cre; D1R$^{flox/flox}$* 4.5 (+/- 0.8 SD) for cKO (P = 0.385, unpaired t test; Fig 1G). When food intake is normalized to body weight, the cKO mice are hyperphagic compared to controls (unpaired t test, P = 0.0009, Fig 1H). Notably, cKO body weights are much lower than controls: prior to starting CR, the average weight of controls was 23.0g (+/- 3.3 SD) while *Vgat-Cre; D1R$^{flox/flox}$* mice weighed only 17.9g (+/- 2.9 SD). At all time points prior to and during timed CR, the body weights of *Vgat-Cre; D1R$^{flox/flox}$* mice were reduced to a greater extent when compared to controls (mixed-effects analysis: fixed effects of time P<0.001, genotype P<0.0001, and time x genotype P = 0.485; for pairwise comparisons using Sidak's multiple comparison, P<0.0001 for days -7, 0, 7, 14 and P = 0.0002 for day 21 and P = 0.001 for day 28; Fig 1I). However, when body weight is normalized by dividing each value by the day 0 body weight, it is clear that the response of body weight to timed 60% CR feeding is similar between *Vgat-Cre; D1R$^{flox/flox}$* and control mice, which both lost similar fractions of weight across the experiment (mixed-effects analysis, fixed effects: time P<0.0001, genotype P = 0.26, time x genotype P = 0.77; P>0.05 for all pairwise comparisons; Fig 1J).

Both groups displayed similar waveforms of normalized high activity behaviors (defined as the sum of walking, cuddling, hanging vertically, hanging upside down, and jumping) on day 0, the first day of timed CR feeding (Fig 1K). On day 14 of CR, the duration of nighttime activity appeared prolonged in the *Vgat-Cre; D1R$^{flox/flox}$* mice while the premeal activity was lower amplitude and had a later onset (Fig 1L). On day 28 of timed CR feeding, both groups showed premeal activity but the control group had an increased amplitude of FAA in comparison to *Vgat-Cre; D1R$^{flox/flox}$* mice (Fig 1M). We plotted total activity levels (in seconds), observing that the *Vgat-Cre; D1R$^{flox/flox}$* mice were considerably more active overall than controls: on days 0, 7, 12, and 21 activity was significantly increased (mixed-effects analysis, fixed effects: time P = 0.023, genotype P < 0.0001, time x genotype P = 0.187; for pairwise comparisons P = 0.038 day 0, P < 0.0001 day 7, P = 0.0154 for day 14, P = 0.0196 for day 21). For pre-meal (ZT 4–6) high activity behaviors in seconds, there was a strong effect of time (P<0.0001) but no interaction of genotype (P = 0.565), and a significant effect of time x genotype (P = 0.0269,

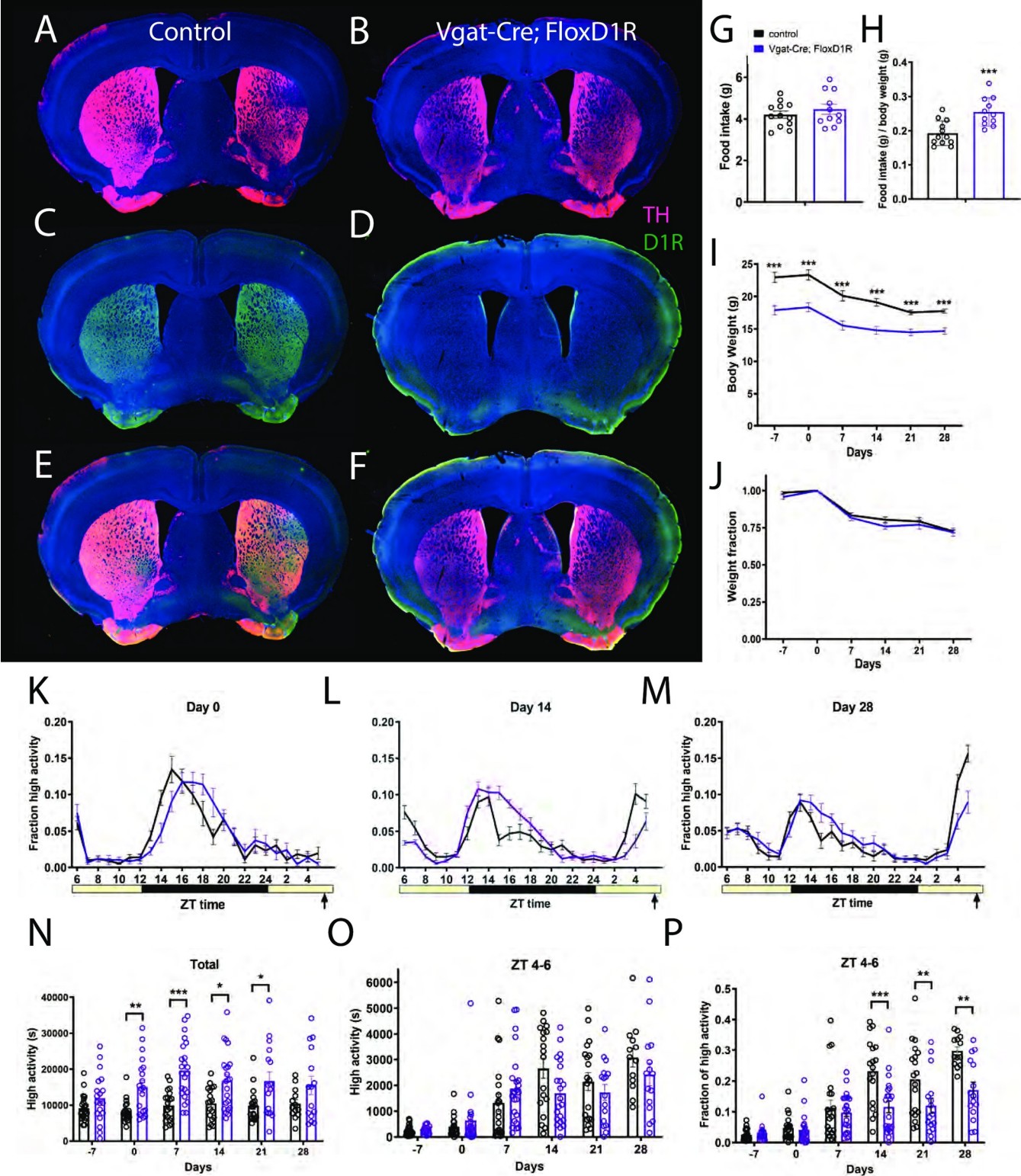

**Fig 1. Immunostaining, food intake, body weight, and circadian behavior of mice with conditional deletion of D1R using vgat-ires-Cre.** Forebrain staining of (A) control TH (pink) and DAPI (blue), (B) Vgat-Cre; FloxD1R TH and DAPI, (C) control D1R (green) and DAPI, (D) Vgat-Cre; FloxD1R D1R and DAPI (E) control merge and (F) Vgat-Cre; FloxD1R merge. (G) Average daily food intake for control (n = 12) and Vgat-Cre; FloxD1R mice (n = 11) in grams and (H) normalized to body weight. (I) Body weight in grams prior to CR and during 4 weeks of CR and (J) expressed as a fraction of Day 0 weight. (K)

Mean ± SEM fraction of high activity behaviors (hanging, jumping, rearing, and walking) day 0 of CR, the first day of scheduled CR feeding. The yellow bar indicates lights on and the gray area indicates lights off. n = 14–23 control and 16–23 cKO for all behavioral measurements. (L) Mean ± SEM fraction of high activity behaviors after 14 days of timed CR feeding. (M) Mean ± SEM fraction of high activity behaviors after 28 days of timed CR feeding. (N) Mean ± SEM total 24 hour high activity data in seconds. (O) Mean ± SEM high activity data in seconds for the 3 h preceding scheduled feeding across the experiment (sum of ZT 5, 6, and 7). (P) Normalized high activity data shown in (O) in the 3 h preceding scheduled feeding across the experiment.

mixed-effects analysis; Fig 1O). However, pairwise comparison of these data yielded no significant differences at any time point (P>0.05). To account for differences in total activity levels between *Vgat-Cre; D1R^flox/flox* and control groups, we divided the ZT 4–6 high activity by the 24-hour total high activity, expressing activity as a fraction of total (Fig 1P). This normalization revealed strong fixed effects of time (P < 0.0001), genotype (P = 0.0029) and time x genotype (P < 0.0001). Pairwise comparisons demonstrated that *Vgat-Cre; D1R^flox/flox* had significantly less FAA as compared to controls on days 14, 21, and 28 of CR (Sidak's multiple comparison, P < 0.0001 day 14, P = 0.0076 day 21, P = 0.024 for day 28). We also noted that by day 28 of timed CR, all control mice showed at least 20% of their activity during ZT 4–6 whereas only a small fraction of the *Vgat-Cre; D1R^flox/flox* mice exceed 20% and many showed very low activity during this preprandial time window.

## Constitutive deletion of D1R in the Drd1^Drago strain moderately reduces food anticipatory activity

Given that *Vgat-Cre; D1R^flox/flox* mice demonstrated more FAA than we had predicted (we expected a near complete loss of FAA), we returned to the constitutive deletion strain *D1R KO^Drago* that we previously demonstrated had a large impairment in FAA [21]. We verified D1R deletion using antibody staining of adult mice; *D1R KO^Drago* mice did not have any D1R staining in the striatum or nucleus accumbens whereas control samples showed abundant staining for D1R (Fig 2A–2F).

Prior to CR, *D1R KO^Drago* mice were individually housed and measured for daily food intakes while on an AL diet. Mean food intake for control animals was 3.6g (+/- 0.8 SD) while *D1R KO^Drago* ate only 3.0g (+/-0.8 SD), which was significantly reduced (Fig 2G, P = 0.020, unpaired t test). However, when food intake is normalized to body weight, the *D1R KO^Drago* mice are hyperphagic compared to controls (P = 0.035, unpaired t test, Fig 2H). Notably, the *D1R KO^Drago* body weights are much lower than controls: prior to starting CR, the average weight of controls was 25.5g (+/- 3.8 SD) while *D1R KO^Drago* mice weighed only 18.3g (+/- 2.8 SD) (Fig 2I). At time points prior to and during timed CR, the body weights of *D1R KO^Drago* mice were less than that of controls (mixed-effects analysis: fixed effects of time P<0.0001, genotype P<0.0001, and time x genotype P<0.001; for pairwise comparisons using Sidak's multiple comparison, P<0.0001 for days -7, 0, 7, 14, 21, and 28; Fig 2I). However, when body weight is normalized by dividing each value by the day 0 body weight, it is clear that the response of body weight to timed 60% CR feeding is similar between *D1R KO^Drago* and control mice, which both lost similar fractions of weight across the experiment (mixed-effects analysis, fixed effects: time P<0.0001, genotype P = 0.32, time x genotype P = 0.013; P>0.05 for all pairwise comparisons; Fig 2J).

Both groups displayed similar waveforms of normalized high activity behaviors on day 0 (Fig 2K). On day 14 of CR, the amplitude of nighttime activity was higher in the *D1R KO^Drago* mice, while their premeal activity was lower than controls (Fig 2L). On day 28 of timed CR feeding, both groups showed premeal activity but the control group had an earlier and stronger increase in the amplitude of FAA in comparison to *D1R KO^Drago* mice (Fig 2M). We plotted the total activity levels (in seconds), observing that the *D1R KO^Drago* mice were

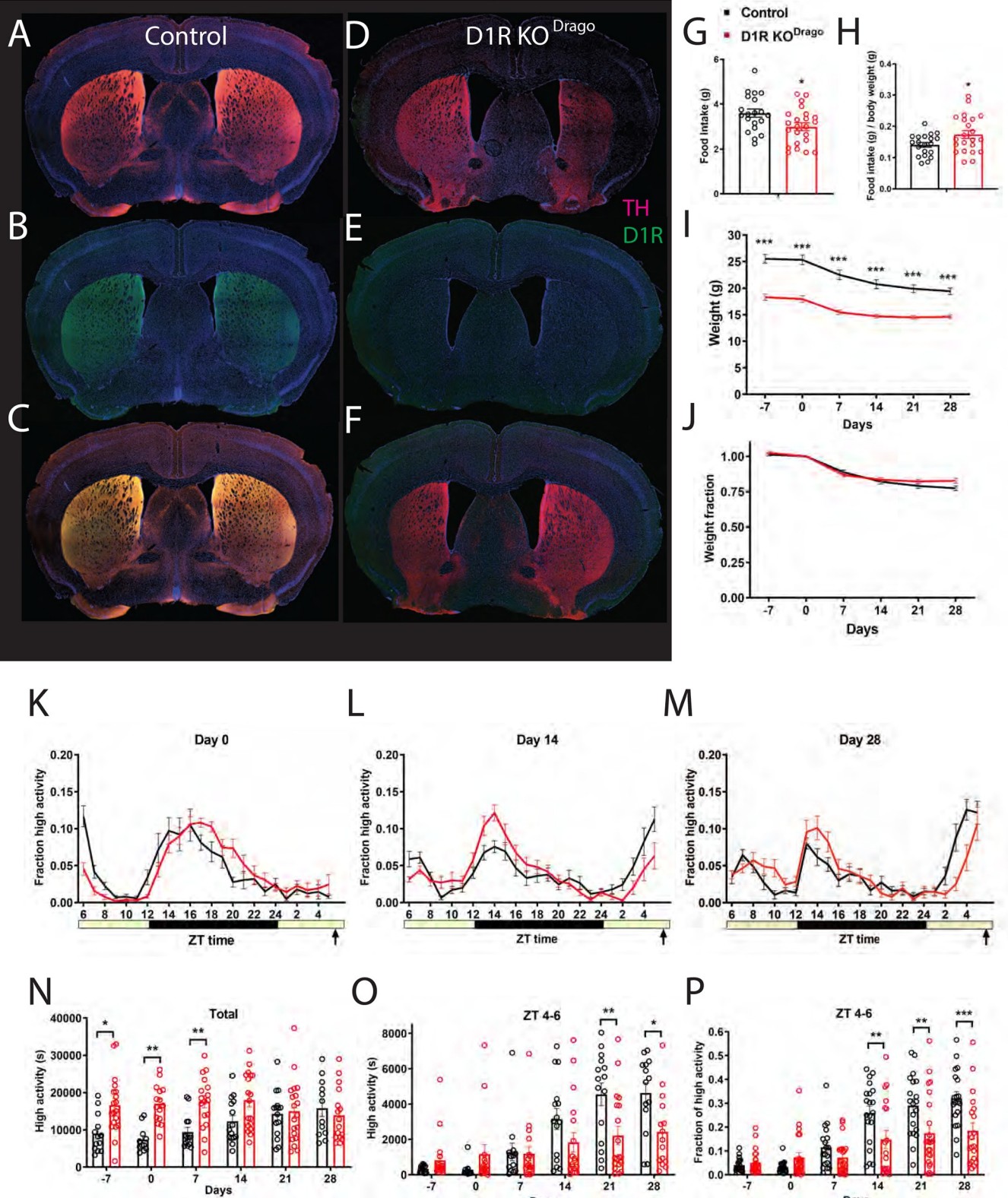

**Fig 2. Immunostaining, food intake, body weight, and circadian behavior of mice with constitutive deletion of D1R (D1R KO$^{Drago}$).** Forebrain staining of (A) control TH (pink) and DAPI (blue), (B) D1R KO$^{Drago}$ TH and DAPI, (C) control D1R (green) and DAPI, (D) D1R KO$^{Drago}$ and DAPI (E) control merge and (F) D1R KO$^{Drago}$ merge. (G) Average daily food intake for control (n = 20) and D1R KO$^{Drago}$ (n = 23) mice in grams and (H) normalized to body weight.

(I) Body weight in grams prior to CR and during 4 weeks of CR and (J) expressed as a fraction of Day 0 weight. (K) Mean ± SEM fraction of high activity behaviors on day 0 of CR. n = 14–23 control and 16–19 KO for all behavioral measurements. (L) Mean ± SEM fraction of high activity behaviors after 14 days of timed CR feeding. (M) Mean ± SEM fraction of high activity behaviors after 28 days of timed CR feeding. (N) Mean ± SEM total 24 hour high activity data in seconds. (O) Mean ± SEM high activity data in seconds for the 3 h preceding scheduled feeding across the experiment. (P) Normalized high activity data shown in (O) in the 3 h preceding scheduled feeding across the experiment.

considerably more active overall on days -7, 0, and 7 but not at later measurements (mixed-effects analysis, fixed effects: time P = 0.401, genotype P = 0.0008, time x genotype P = 0.040; for pairwise comparisons, P = 0.013 day -7, P = 0.0051 day 0, P = 0.0070 for day 7, and P>0.05 days 14, 21, and 28). For pre-meal (ZT 4–6) high activity behaviors in seconds, there was a strong effect of time (P<0.0001) but no interaction of genotype (P = 0.146), and a significant effect of time x genotype (P<0.001, mixed-effects analysis; Fig 2O). Pairwise comparison of these data showed significant differences only on days 21 and 28 of CR (P = 0.038 for day 21, and P = 0.0135 for day 28; P>0.05 at all other measurements, Sidak's multiple comparisons test). To account for differences in total activity levels between *D1R KO$^{Drago}$* and control groups, we divided the ZT 4–6 high activity by the 24-hour total high activity, expressing activity as a fraction of total (Fig 2P). This normalization revealed strong fixed effects of time (P<0.0001), genotype (P = 0.0155) and time x genotype (P< 0.0001, mixed-effects analysis). Pairwise comparisons demonstrated that *D1R KO$^{Drago}$* mice had significantly less FAA as compared to controls on days 14, 21, and 28 of CR (P = 0.0077 day 14, P = 0.0027 day 21, P = 0.002 for day 28, Sidak's multiple comparison). We noted that in contrast to what was observed with *Vgat-Cre; D1R$^{flox/flox}$*, by day 28 of timed CR, several of the *D1R KO$^{Drago}$* mice had ample FAA, exceeding 20 or even 30% of their daily activity during ZT 4–6 (Fig 2P).

## Cryorestored Drd1$^{Drago}$ knockouts have impaired food anticipatory activity

Given that the *Drd1$^{Drago}$* KO strain showed more FAA than expected based on prior results (Gallardo et al., 2014), we speculated that mutations or selection for alleles present in this strain that promote FAA appeared in our line of *D1R KO$^{Drago}$* mice. To test this hypothesis, we obtained the line again from the cryorepository at Jackson lab (Bar Harbor, ME, USA); we denote this line as *D1R KO$^{Drago-cryo}$* to indicate that it is newly established in our colony (Table 1). Prior to CR, mice were individually housed and measured for daily food intakes while on an AL diet. Mean food intake for control animals was 3.8g (+/- 0.4 SD) while *D1R KO$^{Drago-cryo}$* ate 4.4g (+/-0.6 SD) per day, which was significantly increased (Fig 3A, P = 0.012, unpaired t test). When food intake is normalized to body weight, *D1R KO$^{Drago}$* mice are even more obviously hyperphagic compared to controls (P = 0.0001, unpaired t test, Fig 3B). As

**Table 1. Mouse strains used in this study.** A list of the mouse strains along with their abbreviated names used in this study.

| Strain | Allele | Intended deletion | Reference |
|---|---|---|---|
| Drd1$^{tm1Jcd}$/J "*D1R KO$^{Drago}$*" | *D1R* constitutive, null Allele | all | 48 |
| Drd1$^{tm1Jcd}$/J "*D1R KO$^{Drago}$*" cryo-recovered | *D1R* constitutive, null Allele | all | 48 |
| Drd1$^{tm1a(KOMP)Wtsi}$ "*D1R KO$^{KOMP}$*" | *Drd1* constitutive, Null Allele | all | 40 |
| Drd1$^{tm2.1Stl}$/J "*D1R$^{flox/flox}$*" | LoxP flanked *Drd1* allele | all | 43 |
| Tg(Drd1a-Cre)$^{120MXu}$ "*D1-Cre*" | Random integration transgenic; Cre recombinase driven by D1 promoter | D1R in all D1R expressing cells | 44 |
| Slc32a1$^{tm2(cre)Lowl}$ "*Vgat-Cre*" | Cre recombinase driven by vesicular GABA transporter- expressing cells by an internal ribosomal entry site | GABAergic cells | 45 |

**Table 2. Single nucleotide polymorphism in genomic DNA from Drd1<sup>tm1Jcd</sup> and Drd1<sup>tm1e(KOMP)Wtsi</sup> mouse line to assess purity of genetic background.**

| Mouse Strain | Relative to C57BL/6J | Relative to C57BL/6N | Relative to 129S1/SvImJ-FVB/NJ |
|---|---|---|---|
| *Drd1*<sup>tm1Jcd</sup> | 99.65% | -- | 0.3533% |
| *Drd1*<sup>tm1e(KOMP)Wtsi</sup> | 83.34% | 16.66% | -- |

Drd1<sup>tm1Jcd</sup> mice (n = 3) were tested at 144 positions across the genome for the presence of C57BL/6J or 129S1/SvImJ-FVB/NJ alleles. Drd1<sup>tm1e(KOMP)Wtsi</sup> line (n = 4) was tested for the presence C57BL/6J or C57BL/6N strain at 144 positions across the genome. Percentages were in respects to SEM values.

with the other deletion strains, the *D1R KO*<sup>Drago-cryo</sup> body weights are much lower than controls: prior to starting CR, the average weight of controls was 25.7g (+/- 5.8 SD) while *D1R KO*<sup>Drago-cryo</sup> mice weighed only 18.4g (+/- 2.8 SD) (Fig 3C). At time points prior to and during timed CR, the body weights of *D1R KO*<sup>Drago-cryo</sup> mice were reduced as compared to controls at all time points except day 28 (mixed-effects analysis: fixed effects of time P<0.0001, genotype P = 0.0005, and time x genotype P = 0.018; for pairwise comparisons using Sidak's multiple comparison, P<0.0001 for day -7, P = 0.003 for day 0, P = 0.0012 for day 14, P = 0.0036 for Day 14, P = 0.0138 for day 21, P = 0.0585 for 28; Fig 3C). However, when body weight is normalized by dividing each value by the day 0 body weight, it is clear that the response of body weight to timed 60% CR feeding is similar between *D1R KO*<sup>Drago-cryo</sup> and control mice, which both lost similar fractions of weight across the experiment (mixed-effects analysis, fixed effects: time P<0.0001, genotype P = 0.597, time x genotype P = 0.0656; P>0.05 for all pairwise comparisons; Fig 3D).

Both groups displayed similar waveforms of normalized high activity behaviors on day 0, with the *D1R KO*<sup>Drago-cryo</sup> KO showing a wider amplitude of nighttime activity (Fig 3E). On day 14 of CR, the amplitude of nighttime activity was higher in the *D1R KO*<sup>Drago-cryo</sup> mice, while their premeal activity was lower than controls (Fig 3F). On day 28 of timed CR feeding, nocturnal activity continued to be pronounced in *D1R KO*<sup>Drago-cryo</sup> KO mice and they had much less noticeable premeal activity compared to the control group (Fig 3G). We plotted the total activity levels (in seconds), observing that the *D1R KO*<sup>Drago-cryo</sup> mice were more active than controls on days 0 and 7, but not at other measurements (mixed-effects analysis, fixed effects: time P<0.0001, genotype P = 0.257, time x genotype P<0.0001; for pairwise comparisons P = 0.807 day -7, P = 0.0009 day 0, P = 0.0430 for day 7, and P>0.05 days 14, 21, and 28; Fig 3H). For pre-meal (ZT 4–6) high activity behaviors in seconds, there were strong effects of time (P<0.0001), genotype (P = 0.0056), time x genotype (P = 0.0002, mixed-effects analysis; Fig 3I). Pairwise comparison of these data showed significant differences on days 14, 21 and 28 of CR (P = 0.0087 for day 14, P = 0.0330 for Day 21, and P<0.0001 for day 28; P>0.05 at all other measurements, Sidak's multiple comparisons test). To account for differences in total

**Table 3. Observed and expected genotypes for the Tg(Drd1a-Cre)<sup>120MXu</sup> mouse line.**

| F2 Possible Genotypes | Expected | Observed |
|---|---|---|
| WT D1R/flox D1R; D1R-Cre + | 11.75 | 23 |
| WTD1R/flox D1R; D1R-Cre – | 11.75 | 15 |
| Flox D1R/flox D1R; D1R-Cre – | 11.75 | 9 |
| Flox D1R/flox D1R; D1R-Cre + | 11.75 | 0 |

Four possible genotypes were expected as a result of crossing *D1R*<sup>wt/flox</sup>, *D1R-Cre+* and *D1R*<sup>flox/flox</sup>, *D1R-Cre–*mice. Of 47 offspring observed, an expected offspring of 11.75 for each genotype did not match observed, yielding a Chi-square statistic of 24.064 (P<0.0001, Chi-Square).

activity levels between *D1R KO^(Drago-cryo)* and control groups, we divided the ZT 4–6 high activity by the 24-hour total high activity, expressing activity as a fraction of total (Fig 3J). This normalization revealed strong fixed effects of time (P<0.0001), genotype (P = 0.0002) and time x genotype (P = 0.0002, mixed-effects analysis). Pairwise comparisons demonstrated that *D1R KO^(Drago-cryo)* mice had significantly less FAA as compared to controls on days 14, 21, and 28 of CR (P = 0.0060 day 14, P = 0.0016 day 21, P<0.0001 for day 28, Sidak's multiple comparison). We noted that similar to the *D1R KO^(Drago)* mice, by day 28 of timed CR, several *D1R KO^(Drago-cryo)* mice had ample FAA, exceeding 20% of their daily activity during ZT 4–6, while the other half failed to show activity during this time window (Fig 3J).

## A novel constitutive deletion of D1R (Knockout Mouse Project), D1R KO^(KOMP), has delayed acquisition of food anticipatory activity

We next tested a novel constitutive KO of *D1R*, which we term *D1R KO^(KOMP)* (Table 1). This particular deletion strategy was attractive because it used a "knockout first" approach, creating a mutant allele that can later be reactivated by FLP recombinase [40]. In addition, the "flipped" allele can be deleted in a conditional manner by Cre recombinase [40]. As such, we first confirmed that the allele was disrupted by antibody labeling forebrain tissue using D1R antibody (Fig 4A–4F). While TH staining is apparent in both WT and *D1R KO^(KOMP)* forebrain tissue, no D1R was present in the *D1R KO^(KOMP)* forebrain tissue (Fig 4A–4F). In contrast, control samples had ample D1R staining in the striatum and other areas within the forebrain (Fig 4C and 4E).

Daily average food intake was similar between controls, which at 3.2g (+/- 0.7 SD), and *D1R KO^(KOMP)* mice, which ate 3.0g (+/- 0.7 SD) (P = 0.43, unpaired t test; Fig 4G). When food intake is normalized to body weight, there was no difference between groups, unlike for all the other D1R mutants tested (P = 0.62, unpaired t test, Fig 4H). In contrast to the other *D1R* deletion strains described above, the *D1R KO^(KOMP)* mice had only a subtle decrease in body weight (Fig 4I). Prior to starting CR, the average weight of controls was 25.7g (+/- 4.8 SD) grams and *D1R KO^(KOMP)* mice weighed 22.8g (+/- 2.4 SD) (Fig 4I). The only significant difference in body weight occurred at day 0, when the *D1R KO^(KOMP)* mice weighed less than controls (mixed-effects analysis: fixed effects of time P<0.0001, genotype P = 0.0532, and time x genotype P = 0.011; for pairwise comparisons using Sidak's multiple comparison, P = 0.0338 on day 0 and P>0.05 at all other time points). The graph of normalized body weight demonstrates that the response of body weight to timed 60% CR feeding is similar between *D1R KO^(KOMP)* and control mice: both lost similar fractions of weight throughout the experiment (mixed-effects analysis, fixed effects: time P<0.0001, genotype P = 0.607, time x genotype P = 0.374; P>0.05 for all pairwise comparisons; Fig 4J).

In this experiment, all mice displayed similar waveforms of normalized high activity behaviors on day 0 (Fig 4K). On day 14 of CR, the *D1R KO^(KOMP)* mice clearly developed pre-meal activity, but it was of lower amplitude than controls (Fig 4L). By day 28 of timed CR feeding, both groups showed strong pre-meal activity (Fig 4M). We plotted the total activity levels (in seconds), observing that the *D1R KO^(KOMP)* mice were similar to controls and that both groups tended to be less active overall as the experiment progressed (mixed-effects analysis, fixed effects: time P<0.0001, genotype P = 0.550, time x genotype P = 0.029; for pairwise comparisons P>0.05 for all measurements, Sidak's multiple comparison; Fig 4N). For pre-meal (ZT 4–6) high activity behaviors in seconds, there was a strong effect of time (P<0.0001), genotype (P<0.0001), and time x genotype (P<0.0001, mixed-effects analysis; Fig 4O). Pairwise comparison of these data showed significant differences on days 14, 21 and 28 of CR (P = 0.0313 for day 14, P = 0.0019 for day 21, and P = 0.0057 for day 28; P>0.05 at all other measurements,

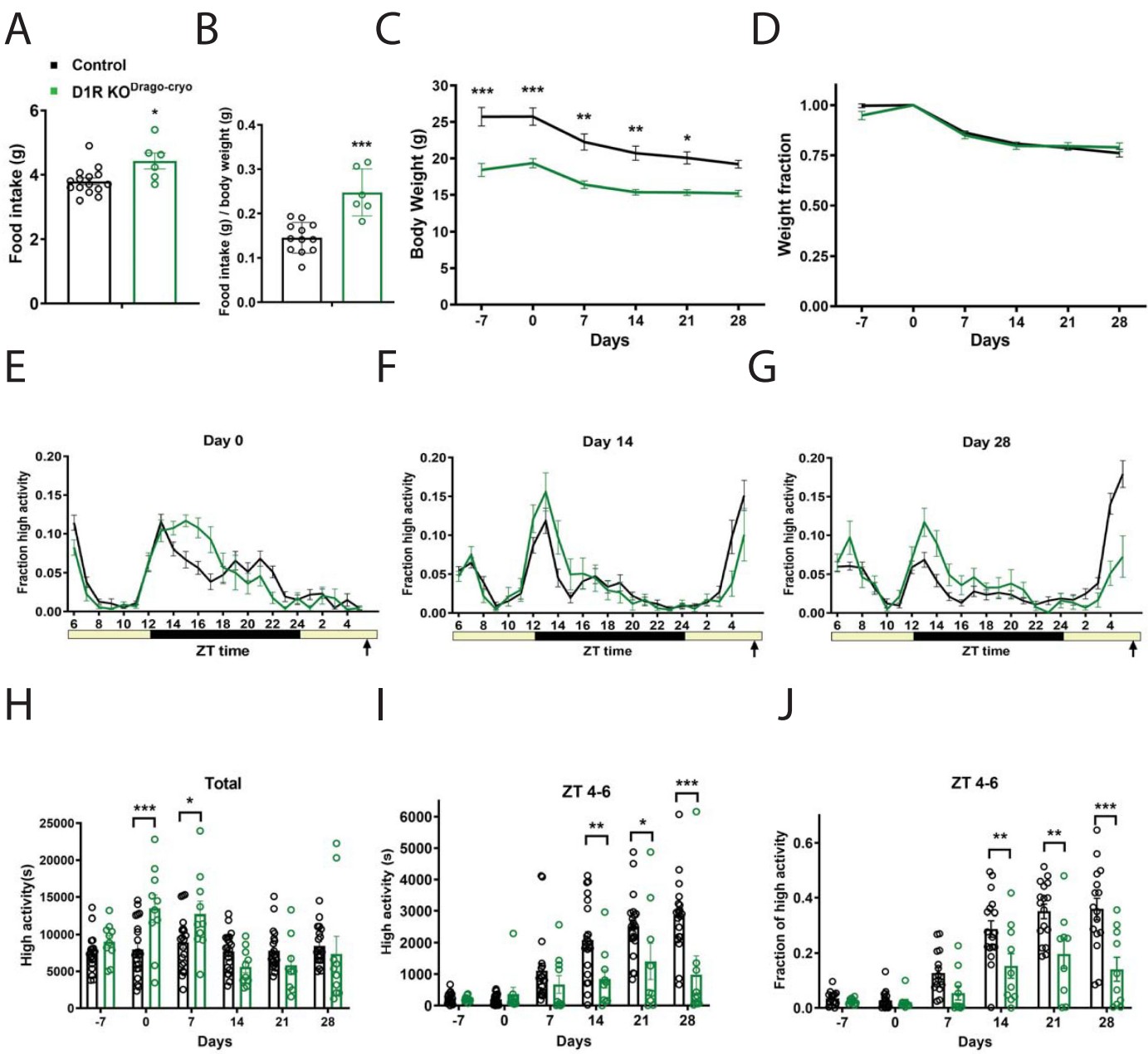

**Fig 3. Food intake, body weight, and circadian behavior of D1R KO^Drago mice from the cryorepository.** (A) Average daily food intake for control (n = 16) and D1R KO^Drago-cryo (n = 8) mice in grams and (B) normalized to body weight. (C) Body weight in grams prior to CR and during 4 weeks of CR and (D) expressed as a fraction of Day 0 weight. (E) Mean ± SEM fraction of high activity behaviors on day 0 of CR. n = 19–21 for control and 9–10 for KO for all behavioral measurements. (F) Mean ± SEM fraction of high activity behaviors after 14 days of timed CR feeding. (G) Mean ± SEM fraction of high activity behaviors after 28 days of timed CR feeding. (H) Mean ± SEM total 24 hour high activity data in seconds. (I) Mean ± SEM high activity data in seconds for the 3 h preceding scheduled feeding across the experiment. (J) Normalized high activity data shown in (I) in the 3 h preceding scheduled feeding across the experiment.

Sidak's multiple comparisons test). To account for differences in total activity levels between *D1R KO^KOMP* and control groups, we assessed activity as a fraction of total (Fig 4P), which revealed a strong fixed effect of time (P<0.0001), genotype (P = 0.0003) and time x genotype (P<0.0001, mixed-effects analysis). Pairwise comparisons demonstrated that *D1R KO^KOMP* mice had significantly less FAA as compared to controls on days 7, 14, and 21 of CR

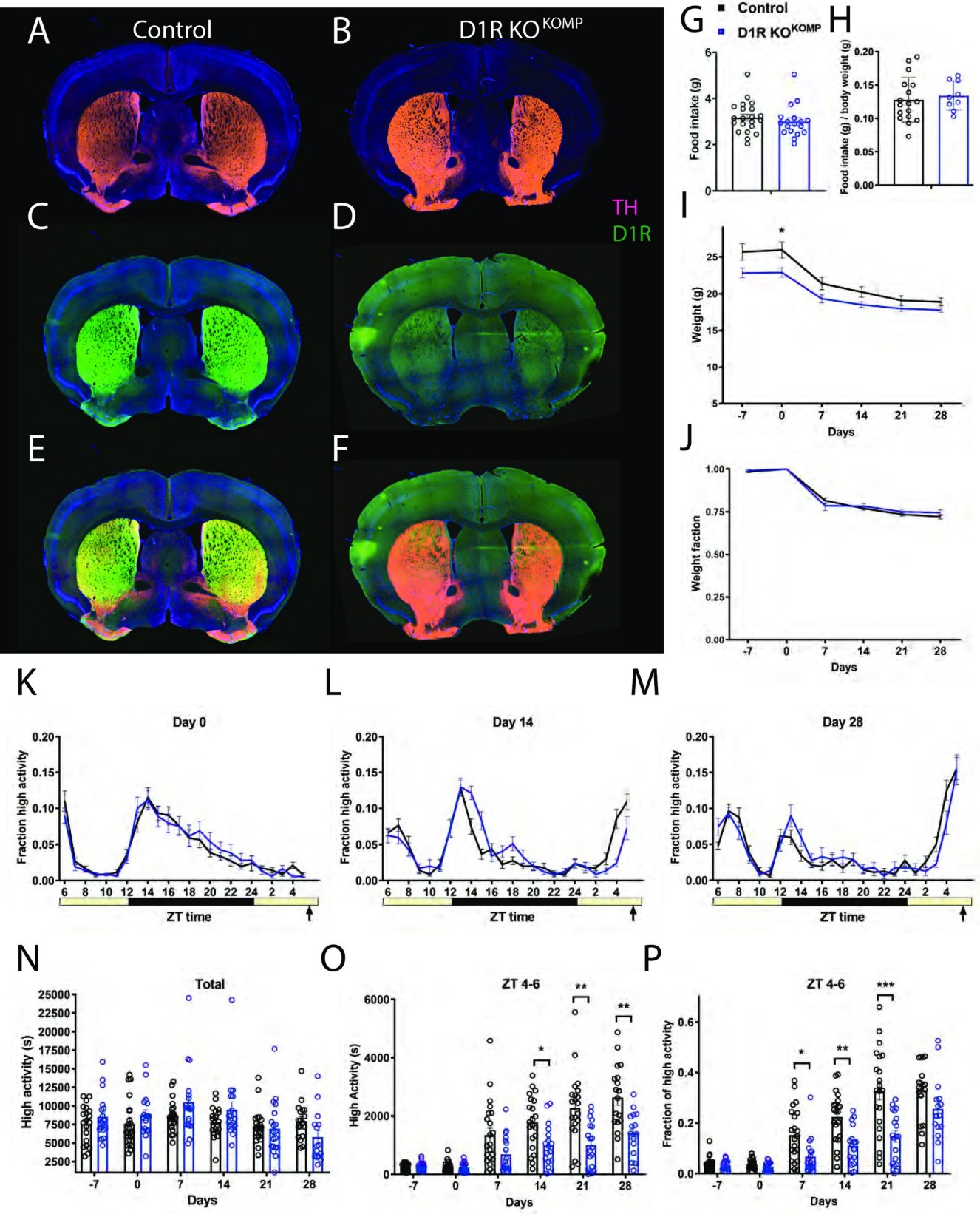

**Fig 4. Immunostaining, food intake, body weight, and circadian behavior of mice with constitutive deletion of D1R (D1R KO[KOMP]).** Forebrain staining of (A) control TH (pink) and DAPI (blue), (B) D1R KO[KOMP] TH and DAPI, (C) control D1R (green) and DAPI, (D) D1R KO[KOMP] and DAPI (E) control merge and (F) D1R KO[KOMP] merge. (G) Average daily food intake for control (n = 21) and D1R KO[KOMP] (n = 18) mice in grams and (H) normalized to body weight. (I) Body weight in grams prior to CR and during 4 weeks of CR and (J) expressed as a fraction of Day 0 weight. (K) Mean ± SEM fraction of high activity behaviors on day 0 of CR. n = 18–22 for control and 16–19 for KO for all behavioral measurements. (L) Mean ± SEM fraction of high activity behaviors after 14 days of timed CR feeding. (M) Mean ± SEM fraction of high activity behaviors after 28 days of timed CR feeding. (N) Mean ± SEM total 24 hour high activity data in seconds. (O) Mean ± SEM high activity data in seconds for the 3 h preceding scheduled feeding across the experiment (sum of ZT 5, 6, and 7). (P) Normalized high activity data shown in (O) in the 3 h preceding scheduled feeding across the experiment.

(P = 0.0286 day 7, P = 0.0014 day 14, P<0.001 for day 21, Sidak's multiple comparison). However, by day 28 of CR, the normalized FAA was similar between groups (P = 0.152), demonstrating that the *D1R KO[KOMP]* mice eventually reach the same level of circadian entrainment to scheduled feeding as controls.

## Discussion

Previously, we provided evidence that deletion of D1R in mice nearly eliminated behavioral anticipation but not the body temperature increase associated with a daily meal; we proposed that the circuitry sufficient for FAA involved D1R mediated signaling from the SN into the dorsal striatum [21]. More recently, we demonstrated that mice with hypomorphic mutations in *paired-like homeodomain transcription factor 3*, a transcription factor necessary for survival of 95% of SN dopamine neurons, defines a minimal population of dopamine neurons sufficient for FAA and metabolic entertainment to timed feeding [47]. To further resolve the circuitry of D1R-expressing neurons and their role promoting FAA, we created a conditional knockout of D1R in *Vgat-Cre* expressing neurons. This deletion eliminated D1R expression in all brain regions examined and significantly attenuated FAA, but some FAA was still present in the conditional KO mice. This led us to retrace our original findings: we repeated testing of FAA in *D1R KO[Drago]* and observed a less prominent attenuation of FAA than we expected based on prior findings. Similarly, testing of other constitutive knockouts of D1R (*D1R KO[Drago-cryo]* and *D1R KO[KOMP]*) showed an impairment in FAA but also that the behavior was still evident (i.e. D1R promotes FAA but is not necessary for it to occur). In sum, our current study suggests that D1R is a key component of FAA but not essential for it to occur.

We considered methodological issues that could have influenced the outcomes of our study. Firstly, the use of *D1R-Cre* line to delete D1R with its endogenous promoter appeared to affect mice embryonically, such that no experimental mice were produced (Table 3). This was surprising because of the several existing constitutive mutants of this allele, which show only slight decreases in viability, but have clear growth defects [48,49]. We speculate that depending on the timing of the deletion of the *Drd1a* gene, compensatory changes occur in the brain to permit viability. Another methodological issue that may be important in comparing results of food entrainment studies relates to the severity of calorie restriction. For example, LeSateur and colleagues (2018) observed that transgenic over-expression of D2R reduced FAA but only on a very mild restriction, whereas the same mice had ample FAA when temporally restricted to a short feeding time window. In prior studies, we have noted that 80% CR induced much lower FAA as compared to 60% CR [50]. It is possible that if we had increased the food allotment in our studies, that the amount of FAA demonstrated by the various D1R knockout strains may have been lessened.

The dissection of the neural circuitry required for FAA has remained elusive. One study demonstrated a strong role for liver-derived ketones in driving food entrainment, yet the brain region(s) mediating responses to ketones has yet to be identified [18]. Furthermore, it appears that there are several brain regions capable of controlling activity cycles in dopamine-

dependent manners [51]. Future studies using conditional genetic approaches may be able to delineate the contributions of separable dopamine and/or D1R populations in mediating the behavioral expression of FAA and/or the underlying food-entrained oscillator. Furthermore, the recently appreciated connection between dopamine signaling in the SCN and out of phase eating resulting in diet-induced obesity demonstrates that this is a complex area of research that deserves additional study [52].

## Supporting information

**S1 File.**
(XLSX)

## Acknowledgments

We are grateful to David Cun, Elizabeth Hill, and Amanda Ng for assistance with mouse behavioral experiments and to Craig LaMunyon for assistance with confocal microscopy.

## Author Contributions

**Conceptualization:** Dina R. Assali, Andrew D. Steele.

**Data curation:** Dina R. Assali, Michael Sidikpramana, Andrew D. Steele.

**Formal analysis:** Dina R. Assali, Michael Sidikpramana, Jeffrey Falkenstein, Andrew D. Steele.

**Funding acquisition:** Andrew D. Steele.

**Investigation:** Dina R. Assali, Michael Sidikpramana, Andrew P. Villa, Jeffrey Falkenstein, Andrew D. Steele.

**Methodology:** Jeffrey Falkenstein.

**Project administration:** Dina R. Assali, Andrew D. Steele.

**Resources:** Andrew P. Villa, Jeffrey Falkenstein.

**Software:** Jeffrey Falkenstein.

**Supervision:** Andrew D. Steele.

**Validation:** Andrew P. Villa.

**Writing – original draft:** Dina R. Assali, Andrew D. Steele.

**Writing – review & editing:** Dina R. Assali, Andrew P. Villa, Andrew D. Steele.

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
