## [Decision Letter · Decision Letter 0]

8 Dec 2020

PONE-D-20-35295

Type 1 dopamine receptor (D1R)-independent circadian food anticipatory activity in mice

PLOS ONE

Dear Dr. Steele,

Thank you for submitting your manuscript to PLOS ONE. After careful consideration, we feel that it has merit but does not fully meet PLOS ONE’s publication criteria as it currently stands. Therefore, we invite you to submit a revised version of the manuscript that addresses the points raised during the review process.

It is important that, in light of the sex differences that have been observed in FAA, that the authors analyze their data for differences among sexes. 

We look forward to receiving your revised manuscript.

Kind regards,

Paul A. Bartell

Academic Editor

PLOS ONE

Journal Requirements:

2. To comply with PLOS ONE submissions requirements, please provide methods of sacrifice in the Methods section of your manuscript.

3. Please include your tables as part of your main manuscript and remove the individual files. Please note that supplementary tables should be uploaded as separate "supporting information" files.

"Funding was provided by the Whitehall Foundation, California State

Polytechnic University Pomona, CSU Program for Education and Research in

Biotechnology, and the National Institute of General Medical Sciences of the

National Institutes of General Medical Sciences of the National Institutes of Health

under Award Number SC3GM125570."

"National Institute of General Medical Sciences of the National Institutes of General Medical Sciences of the National Institutes of Health under Award Number SC3GM125570 to AS. The content is solely the responsibility of the authors and does not necessarily represent the official views of the National Institutes of Health."

Reviewers' comments:

Reviewer's Responses to Questions

**Comments to the Author**

1. Is the manuscript technically sound, and do the data support the conclusions?

Reviewer #1: Partly

Reviewer #2: Yes

2. Has the statistical analysis been performed appropriately and rigorously? 

Reviewer #1: No

Reviewer #2: Yes

3. Have the authors made all data underlying the findings in their manuscript fully available?

Reviewer #1: Yes

Reviewer #2: Yes

4. Is the manuscript presented in an intelligible fashion and written in standard English?

Reviewer #1: Yes

Reviewer #2: Yes

5. Review Comments to the Author

Reviewer #1: The authors used several gene ablation approaches to study the role of D1 dopamine receptor singaling in food anticipatory activity in mice fed 60% of their average ad lib food intake. In all, their data reveal a small effect of loss of D1DR signaling on the level of activity prior to daily feeding. Although not addressed in the paper, these manipulations appeared not to influence the timing of increased activity. This is nicely written paper attempting to shed new light on a long-standing issue in the study of circadian entrainment by feeding schedules, specifically the mechanisms that mediate the heightened activity observed prior to feeding. Unfortunately, in its present form it falls short of adding significant information on the topic. In particular, all experiments were carried-out in both male and female mice (as they should!). However, the reported results do not distinguish between the sexes and data from males and females are pulled together. This is an important issue given evidence of sex differences in food anticipatory activity, and evidence of an important role of female sex hormones on striatal DA receptor-mediated signaling. To gain better insight into the importance of D1DR signaling in FAA, and possibly to discover sexual dimorphism in their contribution, the authors are encouraged to reanalyze and report the data for males and females separately.

Reviewer #2: This is a follow up study on Gallardo et al. 2014 published by the same group. In the initial study, they found marked reduction in FAA upon global DRD1 KO. Here, the authors now conclude that the reduction is rather moderate suggesting that other dopamine receptors might contribute to FAA. In support of their revised claim, the authors present FAA data on Vgat-Cre FloxD1R, global D1RKO, global D1RKO resurrected from cryopreserved embryos, and a different D1RKO line.

The authors used timed 60% caloric restriction to induce FAA assessed by quantifying video-taped ‘high’ activities (jumping hanging rearing ambulating) during the 2hrs prior food replenishment which occurred at ZT6.

The authors found statistical differences in the absolute and relative amounts of FAA in all 4 models, however the dot plots clearly indicate that a fraction of the KO animals still showed FAA more or less indifferent from controls which is also acknowledged by the authors.

The authors confirm that D1 receptor expression is undetectable in the striatum across all models used, arguing against inefficiencies in gene disruption to explain the ‘milder’ phenotype.

Based on these findings, the authors arrive at the conclusion that DRD1 signalling contributes to FAA manifestation but that other dopamine receptors are likely involved as well.

This work is important as it revises previous findings central to the topic of food anticipation, even though the revision is slight.

Perhaps more importantly, the presented data puts the notion of an involvement of DRD1 in FAA on a stronger footing due to the use of different models reporting largely similar deficits.

Comments:

As the Cryo D1KO showed the strongest phenotype it can be argued that some genetic drift perhaps happened, the authors may want to discuss this point (it seems a bit buried). The other genetic model, is it a transcriptional stop knock in? Some detail would be helpful here, considering that stops can be leaky (although the immuno indeed tells otherwise, at least as far as detectable DRD1 levels go).

Are there any GABA- DRD1 neurons? This should be perhaps addressed in the discussion.

I am not quite sure if the lack of obtaining KO offspring from the D1-Cre floxd1 cross should be mentioned in the abstract. The conclusion that the Cre transgene landed on the D1 encoding chromosome seems most parsimonious given that the D1KO phenotypes described in the literature are quite distant from embryonic lethality. Thus being likely only a technical issue, a brief mentioning in the results should suffice.

“The location of the neurons responsible for the anticipation of food and other related stimuli have been elusive”

Do we know if FAA relies on neurons solely? The FAA generator/mediator might include other cells and tissues….

6. PLOS authors have the option to publish the peer review history of their article (what does this mean?). If published, this will include your full peer review and any attached files.

Reviewer #1: No

Reviewer #2: No

---

## [Editor Report · Decision Letter 1]

28 Jan 2021

Type 1 dopamine receptor (D1R)-independent circadian food anticipatory activity in mice

PONE-D-20-35295R1

Dear Dr. Steele,

We’re pleased to inform you that your manuscript has been judged scientifically suitable for publication and will be formally accepted for publication once it meets all outstanding technical requirements.

Kind regards,

Paul A. Bartell

Academic Editor

PLOS ONE
---

## [Editor Report · Acceptance letter]

29 Jan 2021

PONE-D-20-35295R1 

Type 1 dopamine receptor (D1R)-independent circadian food anticipatory activity in mice 

Dear Dr. Steele:

I'm pleased to inform you that your manuscript has been deemed suitable for publication in PLOS ONE. Congratulations! Your manuscript is now with our production department. 

Kind regards, 

on behalf of

Dr. Paul A. Bartell 

Academic Editor

PLOS ONE